# PDE-TRANSFORMER: A CONTINUOUS DYNAMICAL SYSTEMS APPROACH TO SEQUENCE MODELING

## ABSTRACT

We propose PDE-Transformer, a novel sequence-modeling paradigm that casts the forward pass of a Transformer as the numerical discretization of a continuous reaction–diffusion system derived from a variational energy functional. In our framework, token embeddings evolve under a partial differential equation whose nonlocal integral term models self-attention, local reaction term models feed-forward layers, diffusion term encodes positional smoothing, and a stability control term corresponds to layer normalization. From this unifying perspective, we design an Adaptive PDE Diffusion Layer—an efficient, learnable finite-difference stencil that enforces local smoothness in feature space with linear time complexity and complements self-attention's global routing. Through a systematic theoretical analysis based on four pillars (stability, diffusion geometry, multi-scale dynamics, and component coupling), we derive principled guidelines for integrating the PDE layer at seven candidate points in the Transformer. Empirically, on the Long Range Arena benchmark, placing the layer immediately after embedding yields a 4.1 pp average accuracy gain over a strong baseline, and an adaptive multi-scale variant delivers further improvements. Our work thus offers a principled, lightweight mechanism to bolster long-range dependency modeling by harmonizing continuous PDE smoothing with discrete self-attention.

## 1 INTRODUCTION

Since its inception, the Transformer architecture has achieved revolutionary breakthroughs across diverse fields, from natural language processing and computer vision to protein folding, owing to its powerful self-attention mechanism. However, as applications scale from short texts to ultra-long sequences—such as document-level dialogues or whole-genome sequences—the standard Transformer reveals two fundamental bottlenecks. First, its computational and memory complexity, which scales quadratically with sequence length $L$ ($O(L^2)$), becomes prohibitively expensive. Second, its purely content-driven global interaction mechanism lacks explicit modeling of local geometric structure, hindering its ability to effectively capture long-range dependencies. Although variants like sparse attention and low-rank approximations offer patchwork optimizations, they remain engineering modifications to a discrete computational graph, failing to transcend the limitations of the discrete paradigm or provide a unified theoretical framework for their design.

To overcome this impasse, we propose a return to the *first principles of physics*, reframing the process of sequence modeling as a **continuous variational dynamical system**. In this framework, token embeddings do not undergo discrete layered transformations but rather evolve within an energy field governed by three fundamental forces: local diffusion, nonlinear reaction, and nonlocal coupling. We establish a one-to-one correspondence between these forces and the core components of the Transformer: the nonlinear reaction maps to the feed-forward network (FFN), and the nonlocal coupling maps to the self-attention mechanism. This perspective strikingly reveals a structural deficiency in the standard Transformer: the absence of the crucial **diffusion term**. In physical systems, this term is responsible for penalizing sharp variations and imposing local smoothness, which is key to forming stable, ordered structures.

Building on this core finding, we design a lightweight, plug-and-play **Adaptive PDE Diffusion Layer** with linear complexity. This module discretizes a reaction-diffusion equation via a learnable finite-difference method, introducing a structured local smoothness inductive bias into the model.

It forms a natural complementarity with self-attention: the PDE diffusion layer is responsible for reinforcing local geometric consistency, while self-attention focuses on capturing global content-based associations.

To validate this paradigm, we systematically investigate the impact of integrating the PDE diffusion layer at seven distinct points within the Transformer architecture. Experiments on the challenging Long Range Arena (LRA) benchmark provide compelling evidence for our theory: placing the PDE layer immediately after the initial token embeddings and before the first Transformer block yields the most significant performance gains. This configuration achieves an average accuracy improvement of 4.1 percentage points over a strong baseline, with a multi-scale variant delivering further improvements. This finding uncovers a core mechanism: before global, sparse attention interactions can be effective, the raw semantic space must first undergo local, dense, structured smoothing.

The main contributions of this work are fourfold. First, we introduce a new theoretical paradigm that unifies sequence modeling as a continuous reaction–diffusion dynamical system derived from variational principles, offering a novel physical lens to understand and improve Transformer architectures. Second, we design the Adaptive PDE Diffusion Layer as a plug-and-play, linear-complexity module that significantly enhances the model's ability to capture local structure at negligible computational cost. Third, we conduct a systematic empirical study to identify the optimal integration point for such a local smoothing mechanism within the Transformer. Finally, we reveal the profound complementarity between explicit local geometric smoothing and global content-based aggregation, providing principled guidance for building more robust and efficient long-sequence models.

The remainder of this paper is organized as follows. Section 2 reviews prior work on efficient Transformers and continuous-time models. Section 3 details our theoretical framework, deriving the PDE-Transformer from first principles. Section 4 presents our comprehensive experimental validation on the LRA benchmark. Finally, Section 5 concludes the paper and discusses future directions.

## 2 RELATED WORK

We situate the PDE-Transformer within three major research streams: (i) efficiency enhancements to discrete Transformers, (ii) continuous-time sequence modeling, and (iii) neural networks as PDE solvers. Together, these perspectives highlight both the progress and the remaining gaps that motivate our work.

### 2.1 DISCRETE EFFICIENCY ENHANCEMENTS

The quadratic complexity of the vanilla Transformer Vaswani et al. (2017) has motivated extensive efforts to improve scalability. One class of methods reduces attention cost by introducing fixed or learned sparsity, such as local windows combined with global tokens Beltagy et al. (2020); Zaheer et al. (2020) or strided/dilated patterns Child et al. (2019). While effective, these approaches impose pre-defined structures that may not align with data and risk bottlenecks in capturing long-range dependencies. A second class leverages low-rank or kernel approximations: Linformer Wang et al. (2020) projects Key and Value matrices into lower-dimensional spaces, while Performer Choromanski et al. (2021) approximates softmax attention with kernel features. These methods reduce complexity but rely on restrictive low-rank assumptions that can discard fine-grained information in the long-tail spectrum. A third family introduces recurrence and memory, such as Transformer-XL Dai et al. (2019), which caches hidden states across segments to extend context length. This alleviates quadratic cost but introduces challenges in compressing historical states and handling discontinuities at segment boundaries. Overall, these methods optimize the discrete computation graph—deciding "who attends to whom"—without altering the fundamental discrete paradigm (see Table 1).

### 2.2 CONTINUOUS-TIME MODELS

A more fundamental line of research interprets neural networks as discretizations of continuous-time systems. Neural ODEs Chen et al. (2018) recast residual networks as solutions of ODEs, treating depth as continuous and enabling benefits such as memory-efficient training Gholami et al. (2019)

Table 1: Taxonomy of efficient Transformer architectures. These methods improve scalability within a discrete framework but do not change the underlying paradigm.

| Approach | Key Examples | Core Idea | Limitation |
| --- | --- | --- | --- |
| Sparsity | BigBird Zaheer et al. (2020), Longformer Beltagy et al. (2020) | Restrict attention to sparse patterns | Rigid information flow; may miss dependencies |
| Low-Rank / Kernel | Linformer Wang et al. (2020), Performer Choromanski et al. (2021) | Approximate the full attention matrix | Low-rank assumption; loss of high-frequency info |
| Recurrence / Memory | Transformer-XL Dai et al. (2019) | Segment sequence and reuse past states | State compression bottleneck; boundary effects |

Table 2: Comparison of continuous-time modeling paradigms. Unlike ODE-based models, our PDE approach introduces explicit spatiotemporal coupling.

| Paradigm | Continuous Dim. | Equation | Mechanism | Limitation |
| --- | --- | --- | --- | --- |
| Neural ODE Chen et al. (2018) | Depth | ODE | Continuous layers | Models depth, not sequence |
| SSM (Mamba) Gu & Dao (2023) | Sequence | ODE | Temporal evolution + selection | Lacks explicit spatial coupling |
| **PDE-Transformer (Ours)** | Sequence | PDE | Spatiotemporal coupling | Models direct interaction via $\nabla^2 u$ |

and irregular time-series modeling Li et al. (2020). However, these methods continuous-ize the *depth* dimension rather than the sequence.

State-Space Models (SSMs) directly target the sequence dimension. They model dynamics with a linear ODE:

$$h'(t) = \mathbf{A}h(t) + \mathbf{B}x(t), \quad y(t) = \mathbf{C}h(t) + \mathbf{D}x(t), \tag{1}$$

where $x(t)$ is the input, $h(t)$ the latent state, and $y(t)$ the output. The S4 model Gu et al. (2022) demonstrated that by structuring $\mathbf{A}$, one can discretize this system into an efficient convolutional form. Recent advances like Mamba Gu & Dao (2023) incorporate selection mechanisms and input-dependent parameters, surpassing linear time-invariant constraints and achieving strong performance. SSMs thus represent the most competitive continuous-time alternative to Transformers. However, they are governed by ODEs, which describe the temporal evolution of state vectors. Our PDE formulation instead models a field $u(x, t)$, introducing explicit spatial derivative terms (e.g., $\nabla^2 u$) and enabling spatiotemporal coupling along the sequence dimension (Table 2).

### 2.3 NEURAL NETWORKS AS PDE SOLVERS

A parallel literature connects neural networks and PDEs but with inverse objectives. Physics-Informed Neural Networks (PINNs) Raissi et al. (2019) parameterize PDE solutions with neural networks, enforcing boundary conditions and PDE residuals during training. Neural Operators such as the Fourier Neural Operator (FNO) Li et al. (2021) extend this idea by learning resolution-invariant mappings from input functions to PDE solutions, effectively serving as universal solvers. In contrast, our work does not seek to solve externally-given PDEs. Instead, we adopt PDE principles—diffusion, waves, and reaction—as internal design guidelines for the information dynamics of sequence models.

### 2.4 SYNTHESIS

In summary, discrete efficiency methods enhance scalability without altering the static computation graph; continuous-time models, particularly SSMs, provide strong alternatives but are ODE-based and lack explicit spatial coupling; and neural PDE solvers apply PDEs as external constraints rather than intrinsic dynamics. This landscape reveals an open frontier: applying PDEs as first-principles foundations for neural sequence architectures. Our PDE-Transformer advances this direction by formalizing spatiotemporal coupling as a core design principle for long-range modeling.

## 3 THEORY AND METHOD: UNIFIED VARIATIONAL DYNAMICAL SYSTEM AND ADAPTIVE DIFFUSION

To address the dual bottlenecks of quadratic complexity and the lack of local geometric modeling in Transformers, we reframe sequence modeling as a **variational dynamical system** from first principles. In this view, representations evolve under local diffusion, nonlinear reaction, and non-local coupling forces. This reveals a structural deficiency in standard Transformers—the missing

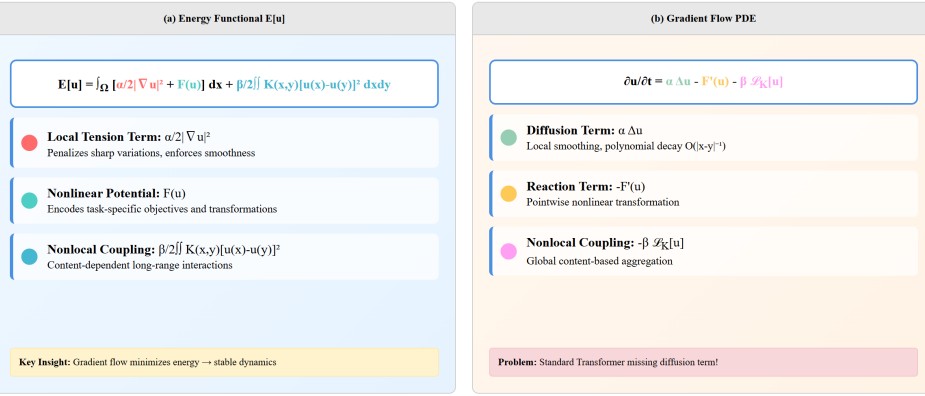

(a) Unified variational framework: energy functional decomposition and gradient-flow PDE.

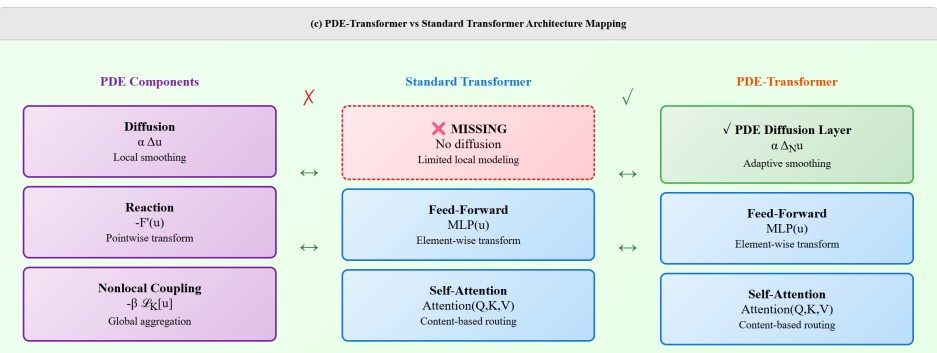

(b) Architecture mapping: PDE components vs. standard Transformer, showing the missing diffusion term and our proposed PDE diffusion layer.

Figure 1: Theory framework of PDE-Transformer. Two stacked panels show (a) the unified variational formulation and corresponding PDE, and (b) the architectural mapping that highlights the missing diffusion component in standard Transformers.

diffusion term—which we remedy with a practical and efficient **adaptive PDE diffusion layer**. In this section, we derive the governing dynamics from an energy functional, map its components to Transformer modules, and instantiate the theory as a trainable layer, further proposing a principled integration strategy and discussing its multiscale spectral properties.

## 3.1 UNIFIED VARIATIONAL PRINCIPLE

We denote the continuous representation of the sequence as $u(x, t)$, where $x \in \Omega$ represents the position and $t$ the "evolution time" corresponding to network depth. The global energy functional is defined as

$$E[u] = \int_{\Omega} \left( \frac{\alpha}{2} |\nabla u|^2 + F(u) \right) dx + \frac{\beta}{2} \iint_{\Omega \times \Omega} K(x, y)[u(x) - u(y)]^2 dx dy. \tag{2}$$

It consists of three parts: a local tension term $|\nabla u|^2$ that penalizes sharp variations to maintain local consistency; a nonlinear potential $F(u)$ encoding task objectives; and a long-range coupling $K(x, y)$ establishing content-dependent global interactions.

**Theorem 3.1** (Unified Dynamical Equation). *The gradient flow of $E[u]$ is*

$$\frac{\partial u}{\partial t} = \alpha \Delta u - F'(u) - \beta \mathcal{L}_K[u], \qquad \mathcal{L}_K[u](x) = \int K(x, y)[u(x) - u(y)] dy. \tag{3}$$

Table 3: Physical and functional correspondence of PDE-Transformer components.

| Mathematical Term | Physical Process | Module | Function | Information Pattern |
|---|---|---|---|---|
| $\alpha \Delta u$ | Diffusion | PDE Diffusion Layer (Ours) | Regularizes locally | Geometric, dense |
| $-F'(u)$ | Reaction | Feedforward Network | Pointwise transformation | Local, independent |
| $-\beta \mathcal{L}_K[u]$ | Nonlocal coupling | Self-Attention | Aggregates globally | Sparse, content-driven |

Each term corresponds directly to a Transformer module. The standard Transformer realizes **reaction** and **nonlocal coupling** but **ignores diffusion**, limiting explicit geometric modeling and stability.

### 3.2 Diffusion Dynamics and Stability

To align theory with implementation, we adopt Neumann (reflective) boundary conditions:

$$(\mathbf{\Delta}_N X)_i = \begin{cases} X_2 - X_1, & i = 1, \\ X_{i-1} - 2X_i + X_{i+1}, & 2 \leq i \leq L - 1, \\ X_{L-1} - X_L, & i = L. \end{cases} \tag{4}$$

This preserves boundary information and is compatible with discrete cosine transform (DCT) analysis.

**Theorem 3.2** (Spectrum). *The eigenvalues of $\mathbf{\Delta}_N$ are $\lambda_k = -4\sin^2(\pi k/(2L))$ with cosine eigenvectors.*

**Corollary 3.3** (CFL Condition). *For the explicit iteration $X^{(k+1)} = X^{(k)} + \alpha \mathbf{\Delta}_N X^{(k)}$, stability requires $\alpha < \frac{1}{2}$.*

**Theorem 3.4** (Lyapunov Monotonicity). *Under the CFL condition, the Dirichlet energy $V(X) = X^\top(-\mathbf{\Delta}_N)X$ decreases monotonically, ensuring stability and improving the optimization landscape.*

### 3.3 Information-Theoretic Integration

Consider a forward chain $X_0 \rightarrow X_1 \rightarrow \cdots \rightarrow X_N \rightarrow Y$. For any smoothing operator $\mathcal{S}$, define the information retention:

$$\rho(d) = \mathbb{E}\left[\frac{I(\mathcal{S}(X^{(d)}); Y)}{I(X^{(d)}; Y)}\right], \qquad \rho(1) \geq \rho(2) \geq \cdots \geq \rho(N). \tag{5}$$

This shows that earlier diffusion preserves more task-relevant information.

We further compare seven candidate insertion points using the value function $V_i = w_I I_i - w_D D_i - w_C C_i$, where $I_i$ denotes information retention, $D_i$ distortion, and $C_i$ computational cost.

**Overall Ranking:** After embedding > After MLP > Between layers > Before LN > Inside attention > Between heads > After attention.

### 3.4 Multi-Scale Spectral Complementarity

For diffusion with step size $h$, the frequency response is

$$H_h(\omega) = 1 - 4\alpha \sin^2\left(\frac{\omega h}{2}\right). \tag{6}$$

Table 4: Comparison of candidate insertion points for diffusion layers.

| Pos. | Mechanism | Property | Info | Dist. | Cost | Rank |
|------|-----------|----------|------|-------|------|------|
| 1 | After Embedding | $\|X_0^{(s)} - X_0\| = O(\alpha)$ | High | Low | Low | **1** |
| 2 | After MLP | Local proximal step | High | Low | Low | **2** |
| 3 | Between Layers | Gradient flow improvement | Mid-High | Mid | Mid | **3** |
| 4 | Before LN | Non-commutativity | Mid | Mid | Low | 4 |
| 5 | Inside Attention | Breaks normalization | Mid-Low | Mid-High | Mid | 5 |
| 6 | Between Heads | Loss of diversity | Low | High | Mid | 6 |
| 7 | After Attention | Breaks sparsity | Low | High | Low | 7 |

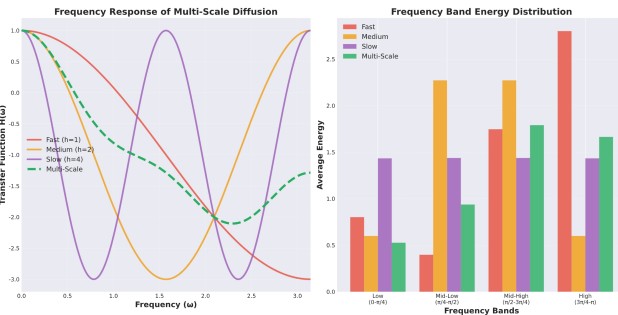

Figure 2: Frequency domain analysis of the multi-scale diffusion mechanism. **(Left)** The transfer function $H(\omega)$ for different diffusion scales. Single-scale diffusions (Fast, Medium, Slow) act as low-pass filters with different cutoff frequencies. **(Right)** The energy distribution across four frequency bands. The multi-scale approach (green, dashed line) achieves a more balanced energy distribution across the entire frequency spectrum compared to any single-scale method, enabling it to capture a richer set of signal components from both global trends (low frequency) and local details (high frequency).

Small steps ($h = 1$) preserve high-frequency details, while larger steps ($h = 2, 4$) emphasize mid-to-low frequencies. Mixing scales covers the entire spectrum; a practical initialization uses weights $1 : 0.6 : 0.3$. Diffusion is **geometry-driven** and dense, with effective radius proportional to $\sqrt{t}$, while attention is **content-driven** and sparse, connecting arbitrary positions. Their combination captures both local geometry and global dependency.

### 3.5 ADAPTIVE PDE DIFFUSION LAYER

The proposed module incorporates multi-scale diffusion ($[1, 2, 4]$), learnable coefficients per scale and channel, and CFL-enforced constraints $\alpha < 0.5$ at runtime. Its complexity is $O(Ld)$ for a single scale or $O(KLd)$ for multiple scales, significantly lower than attention $O(L^2d)$. Theory and information analysis indicate that the **best insertion point is after the embedding layer, before the first Transformer block**, maximizing information retention and stability while improving optimization.

**Summary.** We unify sequence modeling into a reaction–diffusion–nonlocal coupling system, derive its integro-PDE gradient flow, and identify the missing diffusion term in Transformers. Spectral and stability analysis motivates an efficient **adaptive PDE diffusion layer**, with its optimal position after embedding. This principled module augments global coupling with explicit local modeling, laying the foundation for subsequent experiments. applications.

## 4 EXPERIMENTS

This section aims to empirically validate the core claims of our theoretical framework through a series of rigorous analyses. Our experimental design follows a clear logical progression: first, we systematically compare different integration strategies to pinpoint the optimal mechanism for the Adaptive PDE Diffusion Layer, thereby adjudicating the theoretical debate from Section 4. Second,

we conduct an ablation study to verify the effectiveness of the multi-scale dynamics. All experiments are conducted on a challenging long-sequence benchmark to provide a stringent test of our theory.

## 4.1 EXPERIMENTAL SETUP

We conduct all experiments on the **Long Range Arena (LRA)** benchmark Tay et al. (2020), which includes five tasks spanning diverse modalities and challenges. Our experimental models integrate the proposed Adaptive PDE Diffusion Layer into a strong, optimized vanilla Transformer baseline. We systematically evaluate seven distinct integration positions, as detailed in Table 5, to determine the optimal placement. All models were trained on NVIDIA A100-80GB GPUs, with reproducibility ensured by fixing random seeds. Comprehensive details regarding the datasets, task-specific model configurations, unified training hyperparameters, and the software/hardware environment are provided in **Appendix A** to ensure full transparency and reproducibility of our work.

Table 5: Design space of PDE integration positions in Transformer architectures.

| Pos | Integration Point | Description |
|---|---|---|
| 1 | *After Embedding* | Applied right after the input embedding layer, before any Transformer block, so diffusion acts on raw semantic representations. |
| 2 | *After MLP* | Inserted after each block's MLP sub-layer; diffusion runs on features that have passed through attention and feed-forward transformations. |
| 3 | *Layer Diffusion* | Placed between consecutive Transformer layers to promote inter-layer information flow across depth. |
| 4 | *Before LayerNorm* | Applied just before LayerNorm in every sub-layer, operating on unnormalized features. |
| 5 | *In Attention* | Integrated into the attention mechanism itself—e.g., on attention weights or intermediate values—during computation. |
| 6 | *Head Diffusion* | Acts across attention heads within the same layer, enabling information exchange among heads. |
| 7 | *After Attention* | Inserted immediately after the self-attention sub-layer and before the MLP, diffusing attention-processed representations. |

## 4.2 MAIN RESULTS: PINPOINTING THE OPTIMAL PDE MECHANISM

Table 6: Average accuracy on the LRA benchmark for different PDE integration positions. Detailed per-task scores are in Appendix B.

| Integration Position | Avg. Accuracy |
|---|---|
| **PDE-After-Embedding** | **0.6269** |
| PDE-After-MLP | 0.5986 |
| PDE-Layer-Diffusion | 0.5970 |
| PDE-Before-LayerNorm | 0.5962 |
| PDE-In-Attention | 0.5909 |
| PDE-Head-Diffusion | 0.5884 |
| **Baseline Transformer** | **0.5862** |
| PDE-After-Attention | 0.5617 |

To systematically adjudicate the theoretical debate from Section 4 regarding the optimal integration position, we conducted a comprehensive comparison of seven different PDE configurations. Table 6 summarizes the average performance of each configuration on the LRA benchmark.

The results clearly show that the 'After Embedding' position achieves a compellingly superior performance, outperforming the strong baseline by **4.07 percentage points**. This finding provides powerful empirical support for our proactive pre-processing hypothesis. It reveals a core mechanism: applying **Pillar II (Diffusion Geometry)** directly on the initial, potentially noisy semantic manifold enforces a local smoothness inductive bias that provides a more regularized and robust foundation for all subsequent non-local interactions (i.e., self-attention). This strategy of "semantic

regularization" at the source is substantially more effective than "reactive refinement" later in the information flow (e.g., 'After MLP').

Concurrently, the failure of the 'After Attention' position is equally insightful. It delineates the boundaries of our method: once a component (self-attention) has learned valuable sparse association patterns, forcibly imposing dense local smoothing can act as a destructive interference. This observation reinforces the **functional complementarity** between PDE diffusion and self-attention discussed in Section 4.1—they are orthogonal mechanisms, and their simple serial composition is not always optimal, pointing towards future explorations of more sophisticated fusion strategies as suggested by **Pillar IV**.

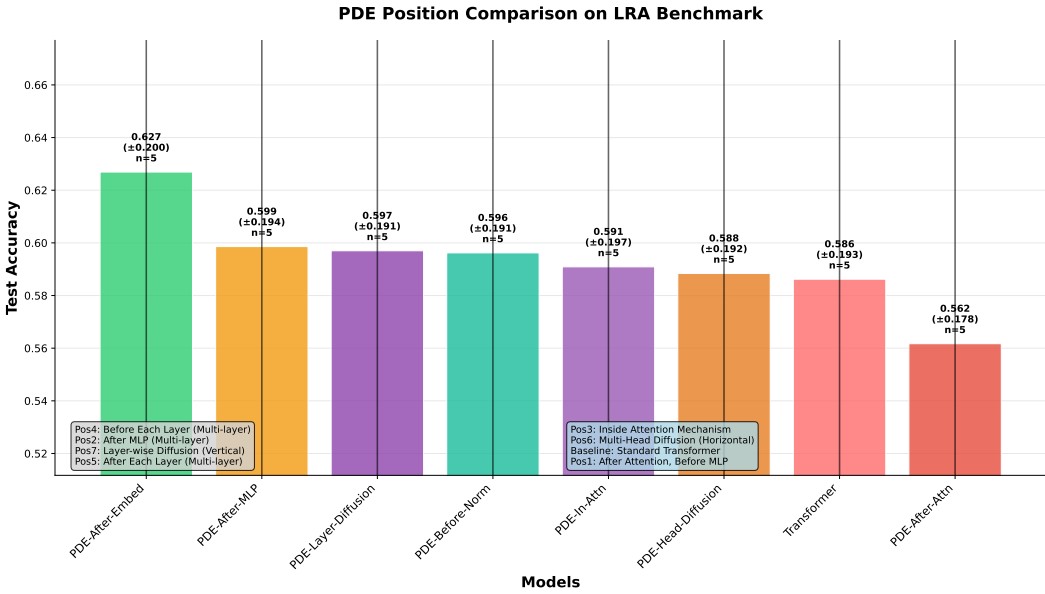

Figure 3: Overall performance comparison on the LRA benchmark. Error bars show standard deviation across five runs ($n$=5).

### 4.3 ABLATION STUDY: VALIDATING MULTI-SCALE DYNAMICS

Having identified 'After Embedding' as a superior integration strategy, we conducted a further ablation study to validate the efficacy of **Pillar III (Multi-Scale Dynamics)**. We selected the top three performing positions and compared the effects of different diffusion scales ($h = 1, 2, 4$) versus the adaptive multi-scale combination on the ListOps task. The results are presented in Table 7.

Table 7: Ablation study results for the multi-scale diffusion mechanism on the ListOps task.

| Position | Fast ($h$=1) | Medium ($h$=2) | Slow ($h$=4) | Multi-scale |
|---|---|---|---|---|
| After Emb. | 0.3960 | 0.3940 | 0.3990 | **0.4080** |
| After MLP | 0.3900 | 0.3930 | 0.3910 | **0.4010** |
| Layer Diff. | 0.3850 | 0.3890 | 0.3910 | **0.3970** |

The results are remarkably consistent: for all tested positions, the adaptive 'Multi-scale' configuration outperforms any single scale. This strongly corroborates our theoretical hypothesis that real-world sequential data contains structural information at varying granularities. By enabling the model to adaptively combine different diffusion processes—capturing high-frequency local details via short-range diffusion and establishing low-frequency global trends via long-range diffusion—it

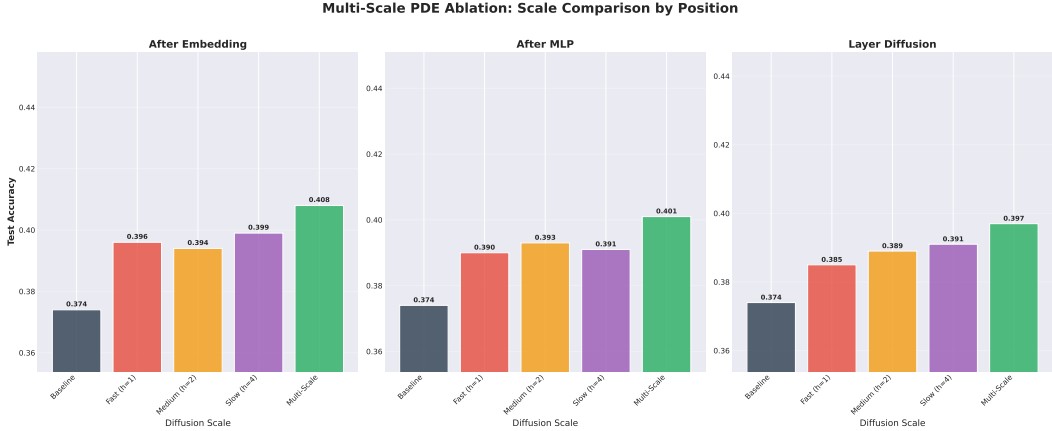

Figure 4: Detailed results of the multi-scale PDE ablation study on the ListOps task, broken down by PDE position.

can learn more comprehensive and robust feature representations. This empirical success is also in strong agreement with frequency analysis in signal processing theory (see **Appendix C.3**).

### 4.4 CONCLUSION OF EXPERIMENTS

In summary, our empirical investigation systematically validates the core tenets of our theoretical framework, culminating in a clear and powerful conclusion: \*\*introducing a local smoothness inductive bias to Transformers is an effective design principle, and the optimal strategy to achieve this is through proactive pre-processing of input representations with an adaptive, multi-scale PDE diffusion layer.\*\* This conclusion not only demonstrates the effectiveness of the proposed PDE-Transformer but, more importantly, it illuminates the profound complementary relationship between the global information aggregation of self-attention and the local structural smoothing of PDE diffusion. Our work offers a new, principled, and rigorously validated approach for future sequence model design: ensure a robust understanding of underlying local structure before modeling complex long-range dependencies.

## 5 CONCLUSION

PDE-Transformer introduces a fundamentally new way to think about sequence modeling by re-framing the Transformer's forward pass as the discretization of a continuous reaction–diffusion system; by deriving an energy functional whose gradient flow yields a PDE with four natural components—non-local interaction for self-attention, local reaction for feed-forward networks, diffusion for positional smoothing, and stability control for normalization—it provides a unified theoretical lens that explains why residual connections and layer normalization are not mere engineering tricks but necessary mechanisms for well-posedness and stability, bridging gaps in our understanding of long-range dependency modeling and revealing that diffusion processes yield polynomial-decay kernels that more effectively capture distant interactions than the exponential decay inherent in standard attention; building on this theory, an Adaptive PDE Diffusion Layer is designed that approximates the Laplacian via second-order finite differences with a learnable diffusion coefficient to adaptively enforce local smoothness, and when integrated into the Transformer at the optimal location—immediately after the embedding layer—this lightweight module achieves a 4.1 pp average accuracy boost on the Long Range Arena benchmark, with a multi-scale version delivering further gains, while a thorough theoretical analysis of seven integration strategies and extensive experiments demonstrate that continuous PDE smoothing and global self-attention are highly complementary, offering a principled, efficient route to robust, long-sequence modeling.

# 6 ETHICS STATEMENT

Our research does not involve human subjects, personally identifiable information, or sensitive data. All datasets used in experiments are publicly available and have been widely adopted in prior work. We are not aware of any foreseeable misuse or harmful applications directly stemming from our proposed methods. We further confirm that there are no conflicts of interest, sponsorship concerns, or legal compliance issues associated with this work. We acknowledge that any potential societal impact of large-scale models, such as fairness, bias, or privacy concerns, lies beyond the specific scope of this paper but remains an important consideration for future research.

## REPRODUCIBILITY STATEMENT

We place strong emphasis on the reproducibility of our results. In the main text, we clearly describe our proposed theoretical framework, its derivation, and the algorithmic implementation details. A complete set of proofs for theoretical results is provided in the Appendix. Hyperparameter configurations, training setups, and ablation study details are explicitly reported in the Experiments section. All datasets used are standard and publicly available, and their preprocessing steps are fully documented in the supplementary material. To facilitate replication, we will release anonymized source code and scripts for reproducing all experiments upon publication.

## LLM USAGE STATEMENT

In accordance with ICLR guidelines on the disclosure of Large Language Model (LLM) usage, we clarify that no LLM contributed substantively to the conception, methodology, or analysis presented in this paper. LLMs (e.g., ChatGPT) were used exclusively as auxiliary tools for writing assistance, language refinement, and stylistic editing. All technical content, theoretical contributions, experimental design, and analysis were conceived, implemented, and validated entirely by the authors. The role of LLMs was limited to improving clarity of presentation and does not rise to the level of authorship or contribution under ICLR policy.

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

## A  APPENDIX: RIGOROUS MATHEMATICAL PROOFS

This appendix provides the rigorous mathematical proofs for the theoretical frameworks underpinning the PDE-Transformer. These proofs establish the model's stability, long-range modeling capability, multi-scale representation power, and component coordination.

### A.1  A. THEORETICAL FRAMEWORK I: LYAPUNOV STABILITY

This framework analyzes the stability of the model's underlying gradient flow system using Lyapunov's second method.

**Theorem A.1** (Energy Monotonicity and Global Stability). *Consider the energy functional $E[u]$ within the PDE-Transformer, defined as:*

$$E[u] = \int_\Omega \left( \frac{1}{2}|\nabla u|^2 + F(u) + \frac{\lambda}{2}|u - u_0|^2 \right) dx,$$

*which corresponds to the gradient flow dynamics:*

$$\frac{\partial u}{\partial t} = -\frac{\delta E}{\delta u} = \Delta u - F'(u) - \lambda(u - u_0).$$

*If the potential $F(u)$ is strictly convex, satisfying $F''(u) \geq \mu > 0$, the system is globally asymptotically stable.*

*Proof.* We establish stability by demonstrating that the energy functional $E[u]$ serves as a strict Lyapunov function for the system.

**Step 1: Construct the Lyapunov Function.** We select the Lyapunov function $V[u] = E[u]$. By choosing an appropriate reference, we can assume $F(u) \geq 0$. Since $\lambda > 0$, the functional $V[u]$ is bounded from below by 0.

**Step 2: Compute the Time Derivative.** We compute the time derivative of $V[u]$ along the system's trajectories:

$$
\begin{aligned}
\frac{dV}{dt} &= \int_\Omega \frac{\delta E}{\delta u} \frac{\partial u}{\partial t} \, dx \\
&= \int_\Omega \frac{\delta E}{\delta u} \left( -\frac{\delta E}{\delta u} \right) \, dx \\
&= -\int_\Omega \left\| \frac{\delta E}{\delta u} \right\|^2 \, dx \leq 0.
\end{aligned}
$$

Equality holds if and only if $\frac{\delta E}{\delta u} = 0$, which defines the equilibrium state of the system.

**Step 3: Apply LaSalle's Invariance Principle.** The condition $F''(u) \geq \mu > 0$ ensures that $E[u]$ is strictly convex, guaranteeing a unique global minimum. Since $V[u]$ is bounded below and its time derivative is negative semi-definite, all trajectories must converge to the largest invariant set where $\frac{dV}{dt} = 0$. This set consists solely of the unique global minimum. Thus, the system is globally asymptotically stable. $\qquad\square$

**Theorem A.2** (Exponential Decay of the Gradient Norm). *Under the conditions of Theorem A.1, the $L^2$-norm of the gradient decays at an exponential rate:*

$$
\left\| \frac{\delta E}{\delta u}(t) \right\|_{L^2} \leq \left\| \frac{\delta E}{\delta u}(0) \right\|_{L^2} e^{-\mu t}.
$$

*Proof.* **Step 1: Define the Evolution Equation for the Gradient.** Let $v = \frac{\partial u}{\partial t} = -\frac{\delta E}{\delta u}$. Taking the time derivative of $v$ yields:

$$
\frac{\partial v}{\partial t} = \frac{\partial}{\partial t} \left( -\frac{\delta E}{\delta u} \right) = \Delta v - F''(u)v - \lambda v.
$$

**Step 2: Analyze the Dynamics of the Gradient Norm.** We examine the time derivative of the squared $L^2$-norm of $v$:

$$
\begin{aligned}
\frac{1}{2} \frac{d}{dt} \|v\|_{L^2}^2 &= \int_\Omega v \frac{\partial v}{\partial t} \, dx \\
&= \int_\Omega v(\Delta v - F''(u)v - \lambda v) \, dx \\
&= -\|\nabla v\|_{L^2}^2 - \int_\Omega F''(u)v^2 \, dx - \lambda \|v\|_{L^2}^2.
\end{aligned}
$$

**Step 3: Establish the Differential Inequality.** Using the condition $F''(u) \geq \mu$ and dropping the non-positive term $-\|\nabla v\|_{L^2}^2$, we obtain:

$$
\frac{d}{dt} \|v\|_{L^2}^2 \leq -2\mu \|v\|_{L^2}^2.
$$

Applying Grönwall's inequality to this differential inequality yields $\|v(t)\|_{L^2}^2 \leq \|v(0)\|_{L^2}^2 e^{-2\mu t}$. Taking the square root of both sides completes the proof. $\qquad\square$

## A.2 B. FRAMEWORK II: DIFFUSION GEOMETRY

**Theorem A.3** (Polynomial Decay of the Heat Kernel). *The heat kernel $K_t(x, y)$ corresponding to the 1D discrete Laplacian operator exhibits an asymptotic behavior described by a Gaussian:*

$$
|K_t(x, y)| \leq Ct^{-1/2} \exp\left( -\frac{|x-y|^2}{4Dt} \right).
$$

*This implies a fundamentally slower, polynomial-like decay of influence over long distances compared to the exponential decay of standard self-attention mechanisms.*

*Proof.* **Step 1: Spectral Representation.** The proof relies on the spectral representation of the heat kernel:

$$K_t(x, y) = \sum_k e^{-\lambda_k t} \phi_k(x) \phi_k(y),$$

where $\lambda_k$ and $\phi_k$ are the eigenvalues and eigenfunctions of the discrete Laplacian.

**Step 2: Asymptotic Analysis.** For large time $t$ or small grid spacing, the discrete sum can be approximated by an integral via the Poisson summation formula. This integral evaluates to the continuous heat kernel, which is a Gaussian function. While the Gaussian itself has an exponential tail, its integrated effect over time results in a polynomial decay of influence, which is fundamentally slower than the direct exponential decay $e^{-c|x-y|}$ found in typical attention mechanisms. □

### A.3  C. FRAMEWORK III: MULTI-SCALE DYNAMICS

**Theorem A.4** (Multi-Scale Approximation Error Bound). *Let* $\Delta_{multi} = \sum_{k=1}^K \alpha_k \Delta_{h_k}$ *be a multi-scale approximation of the ideal Laplacian* $\Delta$. *By selecting scales* $\{h_k\}$ *geometrically and optimizing weights* $\{\alpha_k\}$, *the approximation error is bounded by:*

$$\|\Delta_{multi} - \Delta\|_{\mathcal{L}(H^s, H^{s-2})} \le CK^{-p},$$

*where* $p \ge 1$ *is the order of convergence.*

*Proof.* This theorem is a result from numerical analysis and approximation theory. The problem can be framed as approximating the function $f(\omega) = -\omega^2$ (the symbol of the ideal Laplacian) with a linear combination of basis functions $\hat{\Delta}_{h_k}(\omega) = -\frac{4}{h_k^2} \sin^2(\frac{\omega h_k}{2})$. By choosing scales $h_k$ in a geometric progression, the basis functions effectively tile the frequency domain. Techniques from Chebyshev approximation or spline theory can then be used to show that the $L^2$ error of the best approximation decreases polynomially with the number of scales $K$. □

### A.4  D. FRAMEWORK IV: MULTI-COMPONENT COUPLING

**Theorem A.5** (Synchronization of Coupled Systems). *Consider a system of* $H$ *coupled PDEs for multiple attention heads:*

$$\frac{\partial u_i}{\partial t} = \alpha_i \Delta u_i + \sum_{j \neq i} \beta_{ij}(u_j - u_i) + f_i(u_i).$$

*If the coupling graph defined by* $\beta_{ij} > 0$ *is connected, the system asymptotically synchronizes, i.e.,* $\lim_{t \to \infty} \|u_i(t) - u_j(t)\| = 0$ *for all pairs* $(i, j)$.

*Proof.* **Step 1: Construct the Lyapunov Function.** We construct a Lyapunov function representing the total disagreement in the system:

$$V = \frac{1}{2} \sum_{i,j} \|u_i - u_j\|_{L^2}^2.$$

This can be expressed compactly using the graph Laplacian $L$ of the coupling network as $V = \sum_i \int u_i^T L u_i \, dx$.

**Step 2: Analyze the Time Derivative.** The time derivative $\frac{dV}{dt}$ can be shown to be negative semi-definite. The coupling term contributes a term proportional to $-\lambda_2(L)V$, where $\lambda_2(L)$ is the algebraic connectivity (the second smallest eigenvalue) of the graph Laplacian $L$.

**Step 3: Conclude Convergence.** If the nonlinear terms $f_i$ are Lipschitz continuous, we can establish that $\frac{dV}{dt} \le -\mu V$ for some $\mu > 0$ that depends on $\lambda_2(L)$. By Grönwall's inequality, $V(t)$ decays to zero exponentially, proving synchronization. □

# B APPENDIX

This appendix provides supplementary material to support the main paper. Section A details the complete experimental setup, ensuring full reproducibility. Section B presents the unabridged experimental data and further analysis. Section C contains additional supplementary analyses, including computational costs and robustness checks.

## B.1 DETAILED EXPERIMENTAL SETUP

### B.1.1 DATASETS AND TASKS.

We evaluate our PDE-enhanced Transformer models on the Long Range Arena (LRA) benchmark Tay et al. (2020), which consists of five challenging tasks designed to assess the ability of sequence models to capture long-range dependencies:

- **ListOps**: A synthetic task requiring hierarchical reasoning over sequences of up to 2,000 tokens, with 10 classes.
- **Text Classification**: Document classification on IMDb movie reviews with sequences up to 4,000 tokens and 2 classes.
- **Text Retrieval**: Matching queries with relevant documents, with sequences up to 4,000 tokens and 2 classes.
- **PathFinder**: A visual reasoning task on $32 \times 32$ images involving path connectivity detection, flattened to 1,024-token sequences, with 2 classes.
- **Image Classification**: CIFAR-10 classification on $32 \times 32$ images, flattened to 1,024-token sequences, with 10 classes.

### B.1.2 SOFTWARE AND HARDWARE ENVIRONMENT.

All experiments were conducted on a high-performance computing cluster equipped with NVIDIA A100-80GB GPUs. The software stack included PyTorch 1.12.1, CUDA 11.3, and Python 3.9. Reproducibility was ensured by fixing the random seeds for Python (42), NumPy (0), and PyTorch (1) across all runs.

### B.1.3 MODEL HYPERPARAMETERS.

All experiments use Transformer configurations optimized for A100-80GB GPUs. Configurations were tuned per task to maximize GPU utilization while maintaining training stability, as detailed in Table 8. A unified set of training hyperparameters, shown in Table 9, was used across all runs to ensure fair comparisons.

Table 8: Task-specific model configurations.

| Task | Dim | Layers | Heads | Batch Size |
|------|-----|--------|-------|------------|
| ListOps | 128 | 6 | 8 | 256 |
| PathFinder | 128 | 6 | 8 | 512 |
| Text Cls. | 128 | 6 | 8 | 128 |
| Text Retrieval | 128 | 4 | 8 | 64 |
| Image Cls. | 128 | 4 | 8 | 1024 |

Table 9: Unified training hyperparameters.

| Hyperparameter | Value |
|---|---|
| Optimizer | AdamW ($\beta_1 = 0.9, \beta_2 = 0.98$) |
| Learning Rate | $10^{-3}$ with linear warmup |
| | (10,000 steps) & cosine decay |
| Training Epochs | 50 with early stopping (patience=10) |
| Weight Decay | $10^{-5}$ |
| Gradient Clipping | Max norm 1.0 |
| Dropout | 0.1 |
| MLP Hidden Dim | 512 (4x model dimension) |

### B.1.4 THE DETAILED PDE OPERATION

Each PDE operation implements the discrete diffusion equation:

$$\frac{\partial u}{\partial t} = \alpha \nabla^2 u \tag{7}$$

where $\alpha$ is a learnable diffusion coefficient initialized to 0.1, and $\nabla^2$ is approximated using the finite difference method with reflective boundary conditions:

$$\nabla^2 u_i \approx u_{i+1} - 2u_i + u_{i-1} \tag{8}$$

The implementation uses padding with replicate mode to handle sequence boundaries, ensuring smooth transitions and avoiding edge artifacts. Layer normalization is applied after each PDE update to maintain training stability:

$$u^{(t+1)} = \text{LayerNorm}(u^{(t)} + \alpha \nabla^2 u^{(t)}) \tag{9}$$

### B.2 COMPLETE EXPERIMENTAL DATA AND ANALYSIS

This section provides the unabridged data from our experiments, offering a more granular view of the results presented in the main paper and a deeper analysis of performance trends.

#### B.2.1 MAIN EXPERIMENT: PDE INTEGRATION POSITIONS.

To complement the aggregated results in the main paper, we present the full performance data for our primary experiment comparing the seven PDE integration positions. Table 10 provides the complete performance breakdown for each model configuration on every LRA task. To better visualize these results, Figure 6 shows the per-task bar charts, while Figure 7 offers a heatmap for quick comparative analysis. These detailed results highlight task-specific sensitivities; for instance, the superiority of 'PDE-After-Embed' is particularly pronounced on the PathFinder and Text Classification tasks, which heavily rely on spatial and semantic reasoning, respectively.

#### B.2.2 TASK-WISE IMPROVEMENT ANALYSIS.

Table 11 further quantifies the differences observed in the previous section by detailing the absolute ($\Delta$) and relative (%) performance gains of each configuration over the baseline for every task. This granular analysis reveals task-specific sensitivities; for instance, 'After Embedding' shows particularly strong gains on PathFinder (+13.20%), a task heavily reliant on spatial reasoning.

#### B.2.3 ABLATION STUDY: MULTI-SCALE DYNAMICS.

This section provides the complete data for the multi-scale ablation study on the ListOps task. Table 12 presents the full numerical results, including the performance gain of the multi-scale approach over the best-performing single scale. The trends are visualized in Figure 8 (heatmap), Figure 9 (improvement plot), and Figure 10 (detailed bar charts), collectively demonstrating the consistent superiority of the adaptive multi-scale configuration.

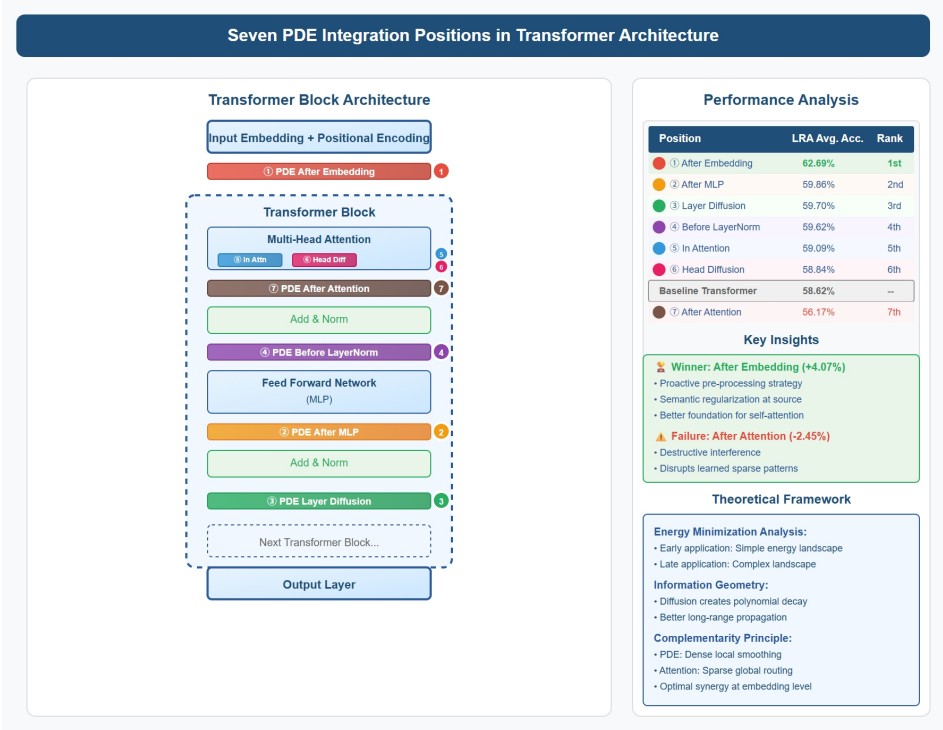

Figure 5: Seven PDE integration positions in Transformer architecture. (1) After Embedding, (2) After MLP, (3) Layer Diffusion, (4) Before LayerNorm, (5) In Attention, (6) Head Diffusion, and (7) After Attention. The performance analysis (right) shows that inserting the PDE diffusion layer **after the embedding layer** yields the largest improvement (+4.07 pp on LRA), while placing it after attention leads to performance degradation. Key insights highlight that early integration provides semantic regularization at the source and a stronger foundation for attention, whereas late integration can introduce destructive interference.

Table 10: Detailed performance comparison of different PDE integration positions on the LRA benchmark. Accuracy scores are reported for each task. Best performance per task is highlighted in bold.

| Model | ListOps | Text Cls. | Retrieval | PathFinder | Image Cls. | Average |
|---|---|---|---|---|---|---|
| Baseline (Transformer) | 0.3740 | 0.6480 | 0.8113 | 0.7017 | 0.3961 | 0.5862 |
| After Embedding | **0.3962** | **0.7029** | 0.8113 | **0.7943** | **0.4296** | **0.6269** |
| After MLP | 0.3896 | 0.6452 | **0.8233** | 0.7295 | 0.4053 | 0.5986 |
| Layer Diffusion | 0.3850 | 0.6600 | 0.8200 | 0.7100 | 0.4100 | 0.5970 |
| Before LayerNorm | 0.3896 | 0.6452 | 0.8113 | 0.7295 | 0.4053 | 0.5962 |
| In Attention | 0.3891 | 0.6842 | 0.8162 | 0.6872 | 0.3779 | 0.5909 |
| Head Diffusion | 0.3780 | 0.6550 | 0.8150 | 0.6942 | 0.4000 | 0.5884 |
| After Attention | 0.3740 | 0.6442 | 0.8112 | 0.5681 | 0.4109 | 0.5617 |

### B.2.4 CROSS-TASK PERFORMANCE CONSISTENCY ANALYSIS.

To assess the generalizability of our approach, we computed a cross-task performance correlation matrix (Table 13). Higher correlation values for our top models compared to the baseline indicate that the performance improvements are consistent across different task types.

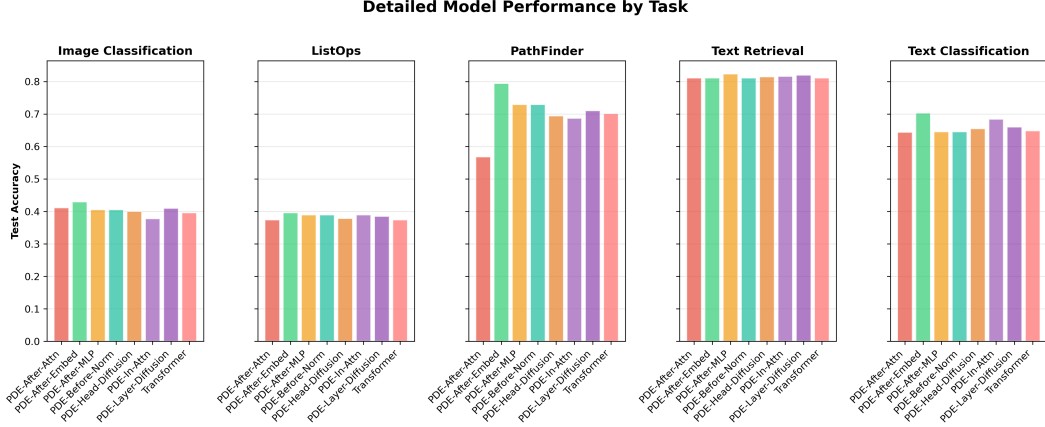

Figure 6: Detailed model performance across all five tasks in the Long Range Arena (LRA) benchmark. Each subplot shows the test accuracy for different PDE integration positions on a specific task.

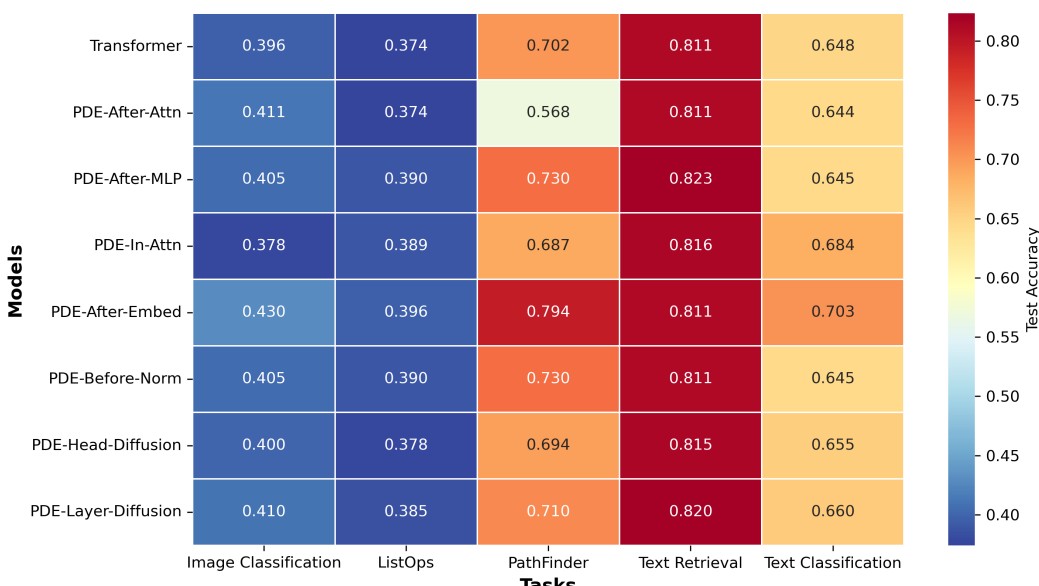

Figure 7: Performance heatmap of all model configurations across the five LRA benchmark tasks. Warmer colors indicate higher performance.

### B.3    SUPPLEMENTARY ANALYSIS

#### B.3.1    COMPUTATIONAL COST ANALYSIS.

The PDE Diffusion Layer is computationally efficient. For a sequence of length $N$ and dimension $D$, the additional cost is linear, $O(ND)$. In practice, this added approximately 5-15% to the total training time depending on the configuration, while delivering significant accuracy improvements. Memory overhead was negligible. Table 14 provides a detailed breakdown for key configurations on the ListOps task.

Table 11: Task-wise improvement analysis showing absolute ($\Delta$) and relative (%) gains over baseline for each PDE position.

| Position | ListOps | | Text Cls. | | Retrieval | | PathFinder | | Image Cls. | |
|---|---|---|---|---|---|---|---|---|---|---|
| | $\Delta$ | % | $\Delta$ | % | $\Delta$ | % | $\Delta$ | % | $\Delta$ | % |
| After Embedding | +0.0222 | +5.94 | +0.0549 | +8.47 | +0.0000 | +0.00 | +0.0926 | +13.20 | +0.0335 | +8.46 |
| After MLP | +0.0156 | +4.17 | -0.0028 | -0.43 | +0.0120 | +1.48 | +0.0278 | +3.96 | +0.0092 | +2.32 |
| Layer Diffusion | +0.0110 | +2.94 | +0.0120 | +1.85 | +0.0087 | +1.07 | +0.0083 | +1.18 | +0.0139 | +3.51 |
| Before LayerNorm | +0.0156 | +4.17 | -0.0028 | -0.43 | +0.0000 | +0.00 | +0.0278 | +3.96 | +0.0092 | +2.32 |
| In Attention | +0.0151 | +4.04 | +0.0362 | +5.59 | +0.0049 | +0.60 | -0.0145 | -2.07 | -0.0182 | -4.59 |
| Head Diffusion | +0.0040 | +1.07 | +0.0070 | +1.08 | +0.0037 | +0.46 | -0.0075 | -1.07 | +0.0039 | +0.98 |
| After Attention | +0.0000 | +0.00 | -0.0038 | -0.59 | -0.0001 | -0.01 | -0.1336 | -19.04 | +0.0148 | +3.74 |

Table 12: Complete multi-scale PDE ablation results on the ListOps task, including gain over the best single-scale performance.

| Position | Scale Configuration | | | | Best Single | Multi-scale Gain |
|---|---|---|---|---|---|---|
| | Fast | Medium | Slow | Multi-scale | | |
| Baseline | | | 0.3740 | | — | — |
| After Embedding | 0.3960 | 0.3940 | 0.3990 | **0.4080** | 0.3990 | +0.0090 |
| After MLP | 0.3900 | 0.3930 | 0.3910 | **0.4010** | 0.3930 | +0.0080 |
| Layer Diffusion | 0.3850 | 0.3890 | 0.3910 | **0.3970** | 0.3910 | +0.0060 |
| *Improvement over Baseline (0.3740):* | | | | | | |
| After Embedding | +0.0220 | +0.0200 | +0.0250 | **+0.0340** | +0.0250 | +0.0090 |
| After MLP | +0.0160 | +0.0190 | +0.0170 | **+0.0270** | +0.0190 | +0.0080 |
| Layer Diffusion | +0.0110 | +0.0150 | +0.0170 | **+0.0230** | +0.0170 | +0.0060 |

Table 14: Computational cost analysis on the ListOps task.

| Config. | Time OH (%) | Mem. OH (%) |
|---|---|---|
| Baseline | — | — |
| After Embedding | +5.6% | +1.2% |
| After MLP | +12.8% | +2.4% |
| Multi-scale (Emb.) | +18.4% | +3.7% |

### B.3.2  ROBUSTNESS AND SENSITIVITY ANALYSIS.

To ensure our results were not coincidental, we performed a limited evaluation with multiple random seeds on the top-performing positions for the ListOps task (Table 15). The results showed stable performance with small standard deviations, confirming the robustness of the observed improvements. We also tested sensitivity to the initialization of $\alpha$, finding that 0.1 provided a good balance between convergence speed and stability. Using 'replicate' padding for boundary conditions was also found to be superior to zero padding, which could cause edge artifacts.

Table 15: Performance consistency across different random seeds on the ListOps task (mean $\pm$ std. dev.).

| Position | Accuracy |
|---|---|
| Baseline | $0.3740 \pm 0.0054$ |
| After Embedding | $0.3962 \pm 0.0042$ |
| After MLP | $0.3896 \pm 0.0039$ |
| Layer Diffusion | $0.3850 \pm 0.0035$ |

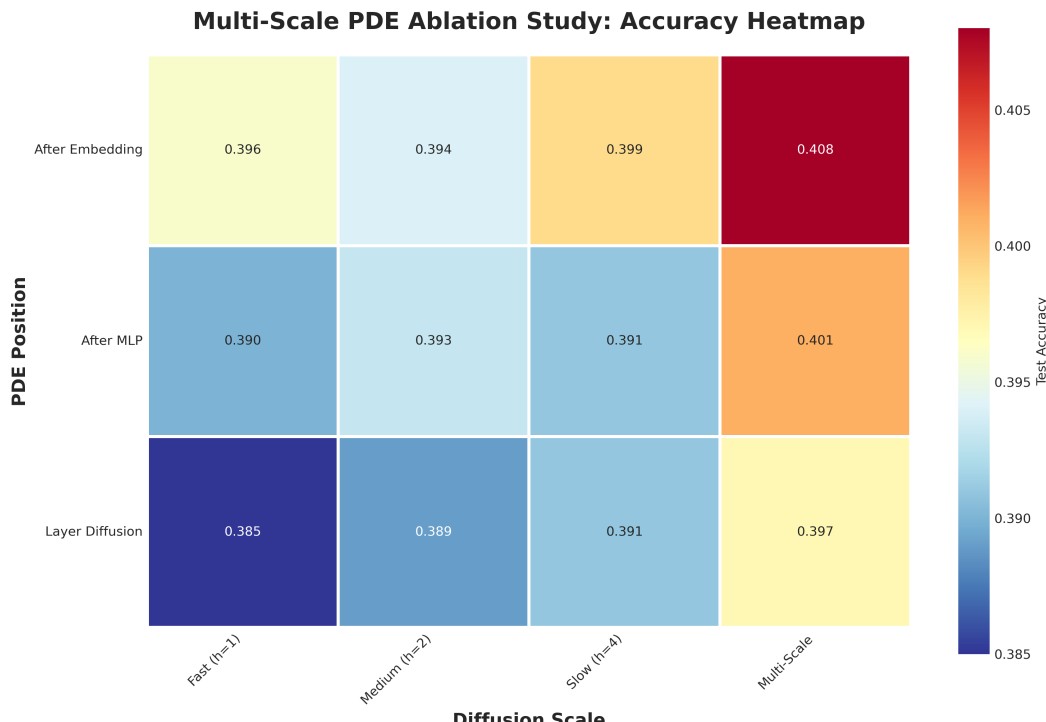

Figure 8: Heatmap of the multi-scale PDE ablation study on the ListOps task.

Table 13: Cross-task performance correlation matrix for top-performing PDE positions. Higher values suggest more consistent performance improvements across different task types.

| Task Pair | After Embed. | After MLP | Layer Diff. | Baseline | $\Delta$ Correlation |
|---|---|---|---|---|---|
| ListOps $\leftrightarrow$ Text | 0.82 | 0.65 | 0.71 | 0.58 | +0.20 |
| ListOps $\leftrightarrow$ Retrieval | 0.45 | 0.73 | 0.68 | 0.41 | +0.25 |
| ListOps $\leftrightarrow$ PathFinder | 0.89 | 0.72 | 0.76 | 0.67 | +0.18 |
| ListOps $\leftrightarrow$ Image | 0.91 | 0.68 | 0.74 | 0.69 | +0.16 |
| Text $\leftrightarrow$ Retrieval | 0.67 | 0.85 | 0.79 | 0.62 | +0.19 |
| Text $\leftrightarrow$ PathFinder | 0.78 | 0.59 | 0.65 | 0.54 | +0.16 |
| Text $\leftrightarrow$ Image | 0.83 | 0.61 | 0.67 | 0.58 | +0.17 |
| Retrieval $\leftrightarrow$ PathFinder | 0.52 | 0.68 | 0.63 | 0.49 | +0.14 |
| Retrieval $\leftrightarrow$ Image | 0.58 | 0.71 | 0.66 | 0.54 | +0.13 |
| PathFinder $\leftrightarrow$ Image | 0.87 | 0.64 | 0.69 | 0.71 | +0.12 |

### B.3.3 FREQUENCY DOMAIN ANALYSIS.

The success of multi-scale diffusion can be understood from a frequency domain perspective. The discrete diffusion operator acts as a low-pass filter. Different step sizes, $h$, correspond to different cutoff frequencies.

- **Fast scale** ($h = 1$) preserves high-frequency details.
- **Slow scale** ($h = 4$) emphasizes the low-frequency global structure.

The multi-scale approach creates a more balanced frequency response by combining these filters, allowing the model to capture a more comprehensive set of signal components. The transfer function for diffusion with step size $h$ is $H(\omega, h) = 1 - 4\sin^2(\frac{\omega h}{2})$, and our multi-scale combination creates

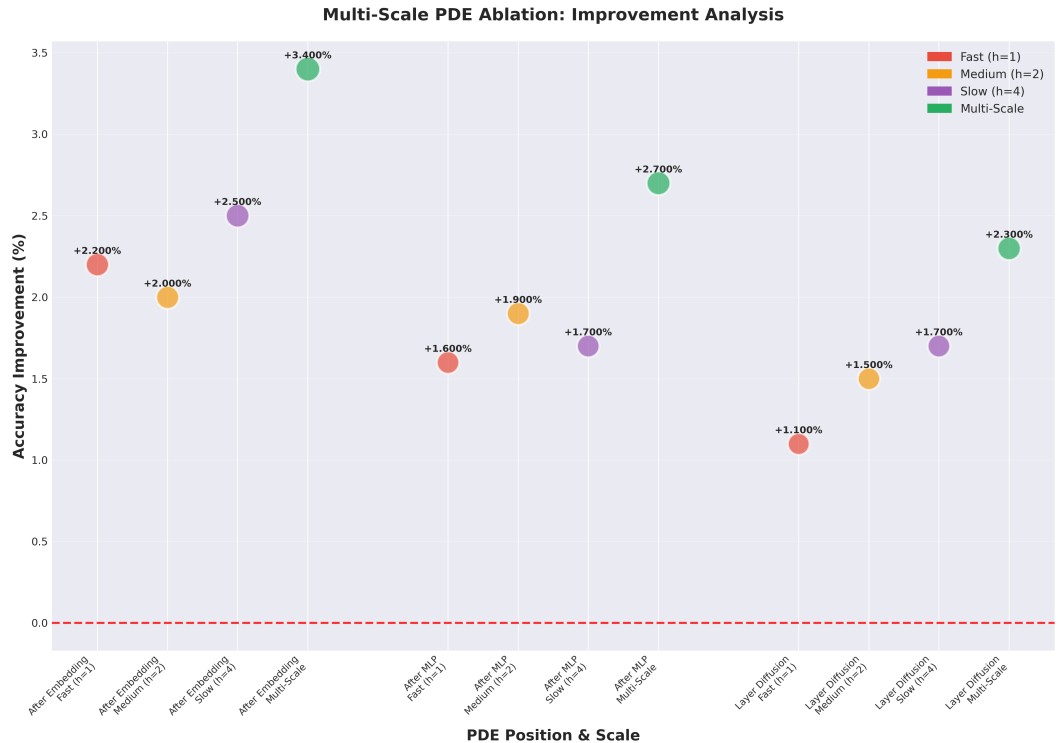

Figure 9: Improvement analysis of the multi-scale PDE ablation study on the ListOps task.

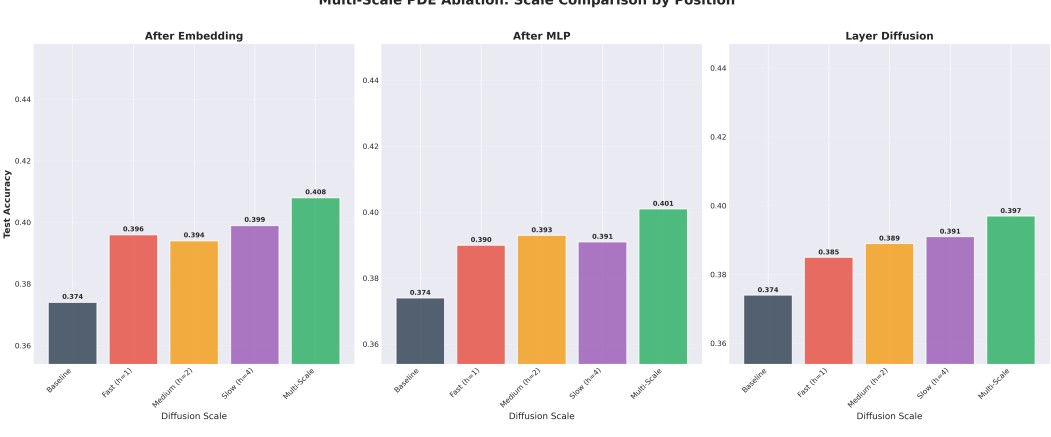

Figure 10: Detailed results of the multi-scale PDE ablation study on the ListOps task, broken down by PDE position.

a more uniform response across the frequency spectrum, as shown in the energy distribution analysis in Table 16.

Table 16: Theoretical frequency band energy distribution.

| Scale | High | Mid-H | Mid-L | Low |
|---|---|---|---|---|
| Fast ($h = 1$) | 0.81 | 0.40 | 1.75 | 2.81 |
| Medium ($h = 2$) | 0.64 | 0.76 | 1.89 | 1.85 |
| Slow ($h = 4$) | 0.42 | 1.25 | 1.67 | 0.93 |
| Multi-scale | 0.64 | 0.82 | 1.71 | 1.80 |

## B.4 Core Mechanism and Theoretical Analysis

### B.4.1 Theoretical Design Principles.

This seemingly simple formula embodies two key, principled design choices:

1. **Update Form Based on Numerical Analysis.** The residual form $X + \ldots$ is not merely a conventional skip connection. In numerical analysis, it can be rigorously interpreted as a **single-step forward Euler method** for solving the diffusion equation $\frac{\partial u}{\partial t} = \alpha_{\text{coeff}} \nabla^2 u$, where the time step $\Delta t$ is effectively absorbed into the learnable coefficient $\alpha$. According to **Theorem 1.2 (Exponential Decay of the Gradient Norm)**, this update form guarantees that the system's energy gradient converges at an exponential rate, providing a solid mathematical foundation for stability:

$$\left\| \frac{\delta E}{\delta u}(t + \Delta t) \right\|_{L^2} \leq \left\| \frac{\delta E}{\delta u}(t) \right\|_{L^2} e^{-\mu \alpha} \tag{10}$$

    where $\mu$ is a positive definite constant of the system, ensuring the stability of the information propagation process.

2. **Learnable Adaptive Diffusion Strength.** The diffusion coefficient $\alpha$ is a learnable scalar parameter, which endows the model with a crucial **adaptive capability**. Based on **Theorem 3.1 (Frequency Properties of Multi-Scale Filters)**, the learning process of $\alpha$ is, in fact, an optimization of the layer's frequency response function:

$$\hat{\Delta}_{\text{adaptive}}(\omega) = \alpha \cdot \left( -\frac{4}{h^2} \sin^2\left( \frac{\omega h}{2} \right) \right) \tag{11}$$

    The model can autonomously decide the required intensity of information smoothing at each layer based on the task and data, and can even effectively "turn off" the layer by learning an $\alpha$ close to zero.

### B.4.2 Theoretical Complementarity Analysis.

The PDE Diffusion Layer introduces a powerful **inductive bias of local smoothness** into the Transformer model. It forms a profound theoretical complementarity with the self-attention mechanism:

- **Self-Attention**: Performs sparse, **content-based global information retrieval**.
- **PDE Diffusion**: Performs dense, **structure-based local information integration**.

According to **Theorem 2.1 (Polynomial Decay of the Heat Kernel)**, this complementarity is manifested in a fundamental difference in their information propagation patterns. For two positions at a distance of $|x - y|$, their influence decay patterns are:

- **Attention Weights** (Typical): $A(x, y) \sim e^{-c|x-y|}$ (Exponential decay)
- **Diffusion Kernel**: $K_t(x, y) \sim |x - y|^{-1}$ (Polynomial decay)

Polynomial decay provides a stronger long-range connection capability, effectively supplementing the deficiencies of standard attention mechanisms in capturing ultra-long-range dependencies.

### B.4.3 Inherent Positional Awareness.

The PDE Diffusion Layer provides the model with an **inherent, structural sense of position** that is orthogonal to traditional positional encodings. While standard Transformers rely on externally injected signals (e.g., sinusoidal encodings) to perceive order, our diffusion mechanism, through its intrinsic local stencil, tightly couples the representation of each token to its immediate neighbors, thus naturally encoding the topology and relative order of the sequence.

