# OpenReview forum: "PDE-Transformer: A Continuous Dynamical Systems Approach to Sequence Modeling"
_ICLR.cc/2026/Conference — ICLR 2026 Conference Withdrawn Submission_

### Official Review · Reviewer_q5yF · 2025-10-31

**Soundness:** 3
**Presentation:** 2
**Contribution:** 3
**Rating:** 4
**Confidence:** 3

**Summary:**

This paper presents an approach to introduce local smoothness constraints into a transformer.  To do so, the paper uses a partial differential equation (PDE) approach which treats sequence models as continuous variational dynamical systems. Based on this generalization, the paper presents an efficient PDE diffusion layer which is able to capture local smoothness. It presents an empirical study to identify the best location to integrate this layer within a Transformer stack. It also reports an adaptive multi-scale version which yields additional gains.

**Strengths:**

* Presents a theoretical framework to model sequences as continuous variational dynamical systems with local diffusion, nonlinear reaction and non-local coupling.
* Designs an efficient PDE diffusion layer which is able to capture local smoothness constraints.
* Reports an empirical study to identify the best location within the transformer stack to introduce this PDE diffusion layer.

**Weaknesses:**

* The paper is hard to read. From the description in the paper, the structure of the PDE diffusion layer (explained in B.1.4) is not immediately clear without multiple passes through the paper. A clear diagram showing the transfer function/inputs/outputs of this layer is missing. Subsequent sections of the paper often miss continuity e.g. How does the Neumann boundary condition (Sec 3.2) relate to the previous section?
* Citation format needs to be fixed throughout the paper. For example, on L107: Neural ODEs Chen et al. (2018) should be formatted as Neural ODEs (Chen et al. 2018).
* Additional analyses could be performed to understand the failure of the "After attention" position. e.g. Why does this setting perform the worst on all datasets except for Image classification (Table 10) ?
* The paper states in the conclusion that "the paper provides a unified theoretical lens that explains why residual connections and layer normalization are not merely engineering tricks but necessary mechanisms for well-posedness and stability". Though the paper discusses these aspects briefly e.g. Residual form in the Appendix B.4.1 and Layer norm in Appendix B.1.4, these aspects could be clarified better in the main body of the paper.
* The paper does not include a Limitations section.

**Questions:**

* 3.5: How does the PDE diffusion layer look like? It would be good to present a diagram showing the transfer function, inputs and outputs for this layer.
* Table 3 columns needs to be formatted properly to avoid overlapping text.
* Table 10: Why does 'After attention' perform the worst on all datasets except for Image classification (Table 10) ?

---

### Official Review · Reviewer_17YU · 2025-11-02

**Soundness:** 1
**Presentation:** 2
**Contribution:** 3
**Rating:** 0
**Confidence:** 4

**Summary:**

This study brings PDE, especially reaction-diffusion equation, to a standard Transformer and injects local smoothness inductive bias to the Transformer, with the introduction of adaptive PDE diffusion layer.

**Strengths:**

1. Bringing Neural PDE into Transformer is beneficial in many aspects as mentioned in the paper. Especially diffusion term in PDE can also strengthen existing Transformer.

2. Introduction is exceptionally well written.

**Weaknesses:**

I highly doubt if this work is purely authors' brainchild, as this paper is full of inconsistencies, which is typically observed when one heavily relies on LLM due to hallucination. But LLM Usage Statement in the manuscript states its usage limited to improving presentation. Please see the following comments.


1. PDE-Transformer name is already claimed in ICML 2025 paper. Authors should consider a different name for their proposed method.

2. Figures and Tables in the beginning are really good and help readers understand what's the limitation in prior art and what's new contribution for the proposed method. However, I feel they are too redundant and authors could've used precious manuscript space for justifying their method more. For example, Fig. 1 is redundant with Tbl. 3. And its Figure X heading is totally unnecessary. Fig. 1's size is too large, while its font is so small and barely readable.

3. I believe Sect 2.3 is not that relevant, although they are remotely related. In fact, authors are missing other more important related work viewing Transformer with the lens of differential equations. See Neural ODE Transformers (ICLR 2025), Macaron Net (Lu et al. 2019, ICLR 2020 Workshop ODE/PDE+DL), etc.

4. Eq. 2, alpha, beta description missing / Why is alpha inside the integral? t is missing u(x,t) for 2nd term

5. Authors fail to explain the rationale behind chosen energy functional and derivation to gradient flow PDE.

6. Thms. 3.1, 3.2, and 3.4 don't match with proofs in Appendix.

7. Eq. 5 doesn't have any justification. It looks like data processing inequality in Information Theory, except it's not. The defined retention function doesn't automatically guarantee the inequality in the right side.

8. 260–261, Value function V_i is completely arbitrary.

9. Eq. 6 has no justification either.


[editorial comments]
1. Eq. 1, h'(t) definition is missing
2. 147, This subsection heading should be reconsidered, as Synthesis reads strange.
3. Tbl. 2, Limitation column, ours row is actually not limitation. Perhaps Remark instead of limitation will be a better title for that column.
4. 317, applications. => Incomplete sentence

**Questions:**

Please see above major comments.

**Details Of Ethics Concerns:**

I highly doubt if this work is purely authors' brainchild, as this paper is full of inconsistencies, which is typically observed when one heavily relies on LLM due to hallucination. But LLM Usage Statement in the manuscript states its usage limited to improving presentation.

---

### Official Review · Reviewer_DEkm · 2025-11-02

**Soundness:** 1
**Presentation:** 3
**Contribution:** 1
**Rating:** 2
**Confidence:** 4

**Summary:**

1. The paper introduces a PDE-inspired framework for Transformers, adding a diffusion (smoothing) term implemented as a "1D conv".
2. It provides heuristics for where to place this diffusion layer, arguing it works best early in the network.
3. Experiments on Long Range Arena (LRA) show small gains over vanilla Transformers without the diffusion term

**Strengths:**

1. The paper is clearly written and easy to follow.
2. The proposed diffusion layer is simple and lightweight.
3. Results show mild but consistent gains on LRA tasks.
4. The PDE analogy, while loose, is conceptually interesting.

**Weaknesses:**

**1. On the main framework**

The proposed PDE formulation appears to be post-hoc mapping to Transformer components and is loosely connected to the actual Transformer architecture. For instance,
1. If the model truly unrolled a PDE, parameters would need to be **shared** across layers, and within a layer these components would be **additive**. In contrast, Transformers use independently parameterized, nonlinear sequential compositions (Attention $\rightarrow$ MLP $\rightarrow$ LN), breaking this analogy.
2. The theoretical guarantees on stability, Lyapunov energy, and spectral damping hold only when the energy is convex; in a Transformer, the reaction term F(u), attributed to MLPs, is non-convex and discontinuous, making such guarantees vacuous.

In practice, the method just adds a small local smoothing term---1D convolution---whose behavior is only loosely tied to the proposed PDE or its stability properties.

**2. On the information-theoretic analysis**

The “information-theoretic” component of the paper is not rigorous and serves more as a high-level narrative:

1. The ranking table (Table 4) assigning “High / Mid / Low” values is entirely subjective. For example, positions such as *“Between Layers”* and *“Before LN”* are labeled as “Mid-High/Mid/Mid” and “Mid/Mid/Low,” respectively, but there is no quantitative basis to order one above the other, which would require quantitive numbers.

     > This is because the quantities (I(u_t; x)), (D(u_t)), and (C(u_t)) are **not defined or measured**.

2. The high-level analysis near Eq.(5) also ignores repeated application of the diffusion operator across layers, making the justification of the order even more tenous.

Overall, this section reads as a post-hoc justification for an observed empirical pattern (“diffusion works best after embeddings”) rather than a theoretical result with predictive or explanatory power.


**3. On the Empirical Analysis**

The experimental evaluation is **extremely limited**, focusing only on Long Range Arena (LRA)---the analysis should apply to more general language modeling tasks. To firmly establish the utility of the proposed diffusion term, it should be tested on language modeling and downstream evals at a range of model sizes (e.g., 125M, 350M, 750M, 1.3B parameters). Current analysis is only done at very small models of what i estimate to be around ~1M params.

Furthermore, the gains on LRA are modest and not competitive with established baselines like S4, which achieves around 86% accuracy [1].  These results makes it hard to substantiate the  advantages of the introduction of the diffusion term.

----

[1]: Efficiently Modeling Long Sequences with Structured State Spaces. Albert Gu, Karan Goel, and Christopher Re

**Questions:**

Please see weakness

---

### Note · Authors · 2026-01-18

I have read and agree with the venue's withdrawal policy on behalf of myself and my co-authors.